# Development and Scalable Production of Newcastle Disease Virus-Vectored Vaccines for Human and Veterinary Use

**DOI:** 10.3390/v14050975

**Published:** 2022-05-06

**Authors:** Julia P. C. Fulber, Amine A. Kamen

**Affiliations:** Viral Vectors and Vaccines Bioprocessing Group, Department of Bioengineering, McGill University, Montreal, QC H3A 0G4, Canada; julia.puppinchavesfulber@mail.mcgill.ca

**Keywords:** Newcastle Disease Virus, viral vaccine bioprocess, bioreactor production, vaccine production platform, clinical trials, COVID-19, SARS-CoV-2, Vero suspension culture

## Abstract

The COVID-19 pandemic has highlighted the need for efficient vaccine platforms that can rapidly be developed and manufactured on a large scale to immunize the population against emerging viruses. Viral-vectored vaccines are prominent vaccine platforms that have been approved for use against the Ebola virus and SARS-CoV-2. The Newcastle Disease Virus is a promising viral vector, as an avian paramyxovirus that infects poultry but is safe for use in humans and other animals. NDV has been extensively studied not only as an oncolytic virus but also a vector for human and veterinary vaccines, with currently ongoing clinical trials for use against SARS-CoV-2. However, there is a gap in NDV research when it comes to process development and scalable manufacturing, which are critical for future approved vaccines. In this review, we summarize the advantages of NDV as a viral vector, describe the steps and limitations to generating recombinant NDV constructs, review the advances in human and veterinary vaccine candidates in pre-clinical and clinical tests, and elaborate on production in embryonated chicken eggs and cell culture. Mainly, we discuss the existing data on NDV propagation from a process development perspective and provide prospects for the next steps necessary to potentially achieve large-scale NDV-vectored vaccine manufacturing.

## 1. Introduction

Infectious diseases have been an important issue throughout human history, with epidemics affecting entire populations, as well as causing losses in livestock [1]. Recently, due to the COVID-19 pandemic, the world turned its attention to viral diseases and vaccine manufacturing. The pipeline to develop and manufacture new vaccines became a global concern, leading to significantly more resources being leveraged into this field [2]. With this increased demand comes a need for the development and optimization of vaccine platforms—technologies that can rapidly be adapted to target an emerging disease, with minimal changes to the manufacturing process [3].

In this context, vaccines based on RNA and viral vectors have shown to be promising options for vaccine platforms. Both avoid the costly biosafety level 3 manipulation of the target pandemic pathogen and are quickly adaptable by modifying the RNA sequence used or the antigen expressed on the viral vector backbone. RNA vaccines stand out for their fast development and ease of production, comprising a straightforward chemical synthesis, with the approved Moderna and Pfizer-BioNTech vaccines playing an essential role in the race for immunization against SARS-CoV-2 worldwide [4]. However, their lack of stability leads to the challenging requirement of frozen storage (lower than −20 °C), which poses issues with transportation, and hinders access in low-income countries and remote regions which lack the cold chain infrastructure [5]. Live virus vaccines, on the other hand, are commonly freeze-dried to be stored between 2 and 8 °C [6].

Viral-vectored vaccines represent a promising alternative with several approved vaccines, including the adenovirus-vectored vaccines used against SARS-CoV-2 (Oxford-Astrazeneca, Johnson & Johnson, Gamaleya and CanSino) [7], and the vectors used against Ebola virus [8] (adenovirus [9], modified Vaccinia Ankara [6,10] and vesicular stomatitis virus [11]). This type of vaccine is highly versatile, with a wide range of human and non-human viruses being studied as vector candidates [12] and the possibility to genetically engineer each vector to modify the surface proteins, generate chimeric strains and select for the desired characteristics, including thermostability [13,14]. This allows researchers to optimize the balance between immunogenicity and safety by modulating virulence and evading pre-existing immunity. It is also possible to design unique vaccination strategies, such as co-expressing different antigens, for a more robust response or even generating a bivalent vector that provides immunization against multiple pathogens [15,16].

Newcastle Disease Virus (NDV) is an avian virus that has been extensively researched as an oncolytic virus, with a long history of clinical trials for this application [17]. Due to host range restriction, it is not pathogenic in humans, which avoids the issue of pre-existing immunity in the population [1]. As such, it is an ideal candidate for a vaccine vector in terms of safety and immunogenicity, and has been implemented in several studies targeting human and veterinary diseases [16]. These vaccine candidates currently rely on the well-established production process in embryonated chicken eggs (ECEs), which can be a cost-effective way of using existing facilities that produce influenza vaccines to produce large quantities of doses [18]. However, there are very few studies exploring the production of NDV in cell culture, which could be scaled to bioreactor production facilities. Cell culture-based processes provide several advantages over production in eggs: they avoid issues with allergens, eliminate dependence on chicken egg supply, and allow greater control over each operation parameter, which leads to more reproducibility, scalability and optimization of the process [3,19].

This review elaborates on the potential of NDV as a viral vector based on its fundamental biology, outlines the recommendations and limitations for designing recombinant NDV constructs, and summarizes the history of developed vaccine candidates for human and veterinary use, including the strains and routes of administration used, as well as highlighting innovative strategies. Lastly, we discuss what has been done in terms of viral propagation and process development, and how this could potentially be leveraged for scalable manufacturing in bioreactors.

## 2. Characteristics of NDV as a Viral Vector

The Newcastle Disease Virus is an avian paramyxovirus that stands out as a promising viral vector based on many advantageous characteristics. Although NDV poses a concern in the poultry industry for the neurological and respiratory disease it can cause among chicken [1], this virus typically does not lead to disease in humans [20]. Only a few cases of conjunctivitis have been reported among those who work closely with poultry or virus samples [21,22,23,24]. This avoids the issue of pre-existing immunity in the population that can arise when using widely spread human viruses as viral vectors [12], such as highly seroprevalent adenoviruses [25]. In addition, NDV replicates efficiently in the respiratory tract, enabling it to be a vector for intranasal vaccines [20,26,27,28,29,30,31,32,33]. This type of vaccine generates both mucosal and systemic immunity against the target disease, which is especially useful for vaccines against highly contagious respiratory diseases, including SARS-CoV-2 [2,33].

Another key aspect of this viral vector is safety. NDV strains are classified into three different pathotypes based on their level of virulence in chicken: lentogenic, mesogenic and velogenic. Lentogenic strains, such as LaSota or B1, show the lowest virulence in chicken [1] and have an extensive documented history of being safe in humans, as has been shown in clinical trials using NDV as an oncolytic agent [34,35]. More detailed information on such trials can be found in the most recent reviews on the oncolytic application of NDV [17,36,37]. Mesogenic and velogenic strains, on the other hand, are typically not used as vaccine vectors due to their virulence in chicken [1] and their status as a “Select Agent” in the United States [13]. As an RNA virus with cytoplasmic replication, NDV also poses very low risk when it comes to the chance of recombination with the host’s DNA, and it has been suggested to lack gene exchange with other viruses [35].

Aside from safety and efficacy, NDV also offers versatility when it comes to production processes. This virus has been produced extensively in embryonated chicken eggs [31,38] and has also shown the capacity of infecting continuous cell lines such as HEK293 [3], Vero [3,20,39], DF-1 [20,38,40], MDCK [41] and HeLa [42]. Although there are very few studies focusing on process development for NDV, it shows the potential to be produced in either egg-based or cell culture-based processes depending on which production facilities are available and which strategies are adopted.

## 3. Designing and Generating Recombinant NDV

NDV contains an RNA genome which is single-stranded, negative-sense and non-segmented, ranging from 15,186 to 15,198 nucleotides in length [43]. The genome comprises six transcriptional units, encoding a nucleocapsid protein (N), a phosphoprotein (P), a matrix protein (M), a fusion protein (F), a hemagglutinin-neuraminidase protein (HN) and a large polymerase protein (L) [1]. In addition, the RNA of the P gene can be edited to generate the V or W proteins, which are non-structural and generally associated with modulating the avian host’s immune response [44]. Techniques for generating recombinant NDV constructs are already well established, with protocols following the general steps of: antigenomic plasmid construction, transfection, rescue and amplification [45].

It is important to consider the limitations and recommendations for recombinant NDV when designing the antigenome plasmid. As a non-segmented genome, large increases in genome length can impair virus replication, and it has been suggested that the maximum transgene size tolerated by NDV is around 3 kb [46,47] or 5 kb [27,36]. NDV constructs bearing genes encoding the SARS-CoV S protein [28] and the SARS-CoV-2 S protein [27], both around 3.8 kb in size, have been successfully generated, demonstrating this vector’s capacity for inserts in this size range. NDV has also been shown to tolerate multiple transgenes [48,49,50,51,52,53,54,55,56,57,58] using different co-expression strategies [59]. Each foreign gene must be flanked by untranslated regions (UTRs) of NDV genes called gene start (GS) and gene end (GE) sequences (Figure 1). The level of transgene expression varies according to the GS and GE, with the highest expression achieved using UTRs from the M and F genes [60]. Any GS or GE-like sequences within the transgene should be removed through silent mutagenesis [61].

Transgenes are typically inserted in the optimal site between the P and M genes (Figure 1), although other sites can also be used [62], and must follow the “rule of six,” which determines that the genome length should be an even multiple of six. This is important for efficient virus replication, as each nucleocapsid protein monomer covers around six nucleotides, so this rule ensures that the nucleotide sequence can be completely encapsidated [43].

For rescue, the antigenome plasmid containing T7 promoter and terminator sequences must be co-transfected with plasmids expressing the N, P and L genes [45]. The T7 DNA-dependent RNA polymerase is commonly introduced by: (i) infecting the cell with a recombinant virus, such as Modified Vaccinia Ankara; (ii) using a stable cell line expressing this polymerase; or (iii) introducing an additional plasmid expressing the T7 polymerase [63]. After rescue, the virus is typically amplified in embryonated chicken eggs [45] or permissive cell lines [38]. Once amplified, the infectious stock of recombinant virus is ready to be used for further production by infecting embryonated chicken eggs or cell culture.

## 4. NDV-Vectored Vaccines

### 4.1. Vaccines for Human Use

NDV has been explored as a vector for human vaccines over the past two decades, with nearly 30 published studies assessing these vaccine candidates in animals (Table 1). Other reviews have listed these studies previously [1,15,64], and an updated table is provided below.

NDV-vectored vaccine candidates have been developed for a range of pathogens, including HIV, EBOV and, predominantly, respiratory viruses such as influenza, SARS-CoV and SARS-CoV-2 (Table 1). Most of the vaccine candidates employed recombinant lentogenic NDV strains as an intranasal live vectored vaccine. The LaSota strain is predominantly used, although other lentogenic strains such as Hitchner B1 and VG/GA, as well as the mesogenic strain Beaudette C (BC), are also present. Interestingly, a few vaccine candidates use chimeric NDV strains, such as the LaSota/VF. This strain is based on a LaSota backbone with the BC strain F protein cleavage sequence, which slightly increases the virulence in birds but allows the virus to replicate in cell culture without the need for added trypsin [28]. In other studies, the mesogenic BC strain was modified by exchanging the ectodomains in the surface glycoproteins F and HN by their equivalents from the LaSota strain [72] or avian paramyxovirus 3 (APMV-3) [13] to reduce virulence and increase safety.

Due to host range restriction, NDV was shown to have attenuated replication in primates, while still generating sufficient mucosal immunity as a respiratory virus [20], demonstrating both safety and immunogenicity. As such, many studies targeting respiratory diseases have implemented the intranasal route of inoculation, including most of the recently developed vaccines targeting SARS-CoV-2 [18,26,27,30,31,76]. Combinations of intranasal and intramuscular doses have also been assessed for several vaccine candidates [30,31,70,76]. Inactivated NDV-vectored vaccines, on the other hand, have been administered exclusively by the intramuscular route, as the inactivated virus can no longer replicate in the mucosal passages [31,75,76,78].

A few studies have used multiple antigens to target the pathogen of concern. Notably, an NDV vector co-expressing the poliovirus P1 and 3CD proteins resulted in the formation of poliovirus viral-like particles (VLPs) in the host cells upon vaccination, which means antigens are presented in a form most similar to the native pathogen while still being safer than live poliovirus vaccines [49,71]. The replication of NDV also serves as a natural adjuvant, increasing the immunogenicity of the vaccine. In another study, NDV was used to co-express the HIV Env and Gag proteins, testing several different orders and positions in the genome [50]. Most constructs in this study also generated VLPs, enhancing the immune response and providing a promising vaccination platform for HIV. Other studies have evaluated the effect of multiple antigens by co-infecting animals with different NDV constructs expressing each antigen separately [69,74], although this approach was less efficient than inoculating a single construct in a study for JEV vaccines [74]. This strategy could avoid issues with slow NDV propagation due to co-expression of multiple transgenes but would require the production of different NDV vectors for the same vaccine, similar to influenza vaccines containing multiple strains [19].

Certain studies have combined different vaccination approaches for a heterologous immunization strategy. One study used an NDV vector expressing the HIV Env (gp160) protein for a priming dose and boosted with purified recombinant proteins (gp120 or gp140), which was the most efficient regimen of vaccination tested [73]. This mixed regimen induced higher magnitude of immune response than the regimen with only NDV-vectored doses, while also providing a longer lasting immune memory in comparison to the purified protein-only regimen. As such, the NDV-vectored prime was important for long-term immunity, while the protein boost enhanced immunogenicity. Another study explored combining different vectors by priming with an adenovirus-vectored vaccine and boosting with an NDV-vectored vaccine, or vice versa [77]. This heterologous regimen induced a more potent and robust response than the homologous alternatives, potentially due to avoiding pre-existing immunity to each vector.

NDV-vectored vaccines have great potential to be used against pandemic diseases, having shown efficient protection against EBOV [13,29,77], SARS-CoV [28] and SARS-CoV-2 [18,26,27,30,31,75,76,78]. Intranasal vaccines against EBOV have been shown to induce neutralizing antibody responses in monkeys [29], guinea pigs [13] and mice [77], although the latter study found more robust responses when mixing adenovirus and NDV-vectored doses as compared to a homologous NDV-vectored regimen. An intranasal vaccine against SARS-CoV [28] also generated a protective antibody response in monkeys and highlighted the need for two doses, as one dose provided insufficient immunogenicity. A similar result was found in a study against SARS-CoV-2 in hamsters [27], in which two doses induced a protective neutralizing response while the single-dose regimen did not significantly reduce viral loads upon infection. Inactivated NDV-vectored SARS-CoV-2 vaccines have also been successful, inducing higher neutralizing responses in mice than a purified protein vaccine [78] and significantly reducing viral loads in a hamster model [75].

Notably, some NDV-vectored SARS-CoV-2 vaccines provided potent protective immunity and reduced viral loads to undetectable amounts on day 4 or 5 post infection, but they did not induce sterilizing immunity, as there were still detectable amounts of virus on day 2 post infection that could potentially lead to shedding [26,27,75]. However, it is considered that widely available vaccines capable of reducing disease severity are highly beneficial in a pandemic, even if sterilizing immunity is not achieved [75].

Although there have been numerous NDV-vectored vaccine candidates tested in animal models (Table 1), only a select few have proceeded to clinical trials. The increased demand for vaccines throughout the COVID-19 pandemic advanced the field, leading to a series of clinical trials for two vaccine candidates (Table 2). A research group based in the Icahn School of Medicine at Mount Sinai (USA) engineered the HexaPro-S (HXP-S) version of the SARS-CoV-2 S protein, which is stabilized in its pre-fusion conformation and anchored in the NDV membrane by containing domains from the NDV F protein [31]. Using the LaSota strain backbone expressing the HXP-S antigen, two vaccine candidates were generated: a live version [18] (“Patria”), in Phase II clinical trials in Mexico, and an inactivated version [75], in Phase I/II clinical trials in Thailand (“HXP-GPOVac”), Vietnam (“COVIVAC”) and Brazil (“ButanVac”) [76]. These vaccines are produced in embryonated chicken eggs (ECEs) in GMP-certified facilities in each country, taking advantage of the existing infrastructure for production of influenza vaccines. The inactivated version uses beta-propiolactone (BPL) for inactivation and CpG 1018 as an adjuvant, while the live version has no adjuvant addition, as the replicative virus is considered self-adjuvanted [76].

Interim results for Phase I clinical trials of the NDV-vectored vaccine against SARS-CoV-2 have been reported using the live version in Mexico [79] and the inactivated version in Thailand [80]. The results showed both versions of the vaccine to be safe in humans. The live version was sufficiently immunogenic at the highest dose tested (10^8^ EID_50_), and all formulations were safe. Interestingly, adequate immunogenicity was only achieved when doses were administered twice intramuscularly or intramuscularly followed by intranasally. The exclusively intranasal regime was found to generate cellular immunity but lacked a robust systemic antibody response. As such, the high-dose formulations with IM-IM and IN-IM routes were chosen for the Phase II trials [79]. As for the inactivated vaccine candidate, the immunogenicity was proportional to the dosage and was not considerably affected by the use of the adjuvant CpG 1018. The mid-dose (3 μg) was considered sufficiently immunogenic and was chosen for the Phase II trials, with and without CpG 1018 [80].

### 4.2. Vaccines for Veterinary Use

NDV has been extensively explored as a vector for veterinary vaccines, with over 60 published studies (Table A1) on vaccine candidates over the past two decades mainly targeting use in poultry or cattle. Other review papers have listed and summarized some of these studies [1,15,16,64,81], and an up-to-date table is provided in the Appendix A (Table A1).

Importantly, as virulent NDV strains cause severe disease in chicken and economic loss, the lentogenic strains are often used to vaccinate chicken for protection against virulent NDV. As such, several studies have implemented a bivalent vaccine approach for poultry in which NDV expresses antigens of another avian virus to immunize chicken against both diseases, including highly pathogenic influenza virus (HPAIV) [41,53,55,82,83,84], avian reovirus (ARV) [85], infectious laryngotracheitis virus (ILTV) [86,87] and fowl adenovirus serotype 2 (FAdV-4) [88]. Notably, certain bivalent vaccines against NDV and HPAIV have been licensed for use in poultry [15,81]. Bivalent NDV vaccines have also been developed for ducks against duck Tembusu virus (DTMUV) [89] and HPAIV H5N1 [90]; for geese against goose parvovirus (GPV) [91] and goose astrovirus (GoAstV) [92]; and for turkeys against avian metapneumovirus (AMPV) [52].

Similar to vaccine candidates for human use, several NDV vaccines expressing multiple antigens have been developed for veterinary use either by co-expressing the antigens within the same vector [51,52,53,54,55,56,57,58] or by inoculating different vectors, each expressing a different antigen, in the same formulation [71,86,93,94]. For the latter, two studies have found that a formulation with only one vector/antigen yielded better results and sufficient protection [86,93], while another study found adequate immune responses regardless of inoculating one or multiple vectors [71].

NDV strains have also been engineered to generate chimeric or modified strains with novel characteristics. Thermostability is a key characteristic for poultry immunization through spraying or drinking water, while also relieving difficulties associated with cold chain requirements [14]. To address this, thermostable NDV strains have been isolated and modified to be used as vectors in avian influenza vaccines [14,95]. Another concern in poultry immunization is pre-existing immunity to NDV [64], which led to the development of chimeric NDV strains in which the native surface glycoproteins (F and HN) were substituted by the corresponding genes from another virus, namely APMV-8 [96] or APMV-2 [97,98]. Chimeric strains have also been developed to modulate virulence by substituting the F and HN proteins of the mesogenic BC strain (partly or completely) with the corresponding proteins from the lentogenic LaSota strain, or by modifying basic residues in the F protein cleavage site [32,99].

## 5. Manufacturing of NDV-Vectored Vaccines

### 5.1. Workflow for Viral Vector Production

Viral vectors are typically produced in ECEs or cell culture, from which they are harvested and purified for vaccine formulation (Figure 2). NDV has been extensively produced in ECEs for poultry vaccination and for pre-clinical studies, with only a few studies propagating this virus in cell lines (Table 1 and Table A1) and, to our knowledge, only four studies producing it in lab-scale bioreactors [3,100,101,102].

For production in ECEs (Figure 2A), eggs must be acquired and incubated at 37 °C prior to inoculation. They are infected between 9 and 11 days old, and are incubated for another 24 h for viral production. ECEs containing dead embryos before 24 h of incubation are discarded, while embryos that die after that timepoint are stored at 4 to 8 °C for several hours before collecting the allantoic fluid to harvest NDV [103].

For production in cell culture (Figure 2B), bioreactors are inoculated with defined medium and the chosen cell line for the cell growth phase. Critical operation parameters such as pH, oxygen concentration, agitation and temperature are kept constant throughout the entire run. Once the required cell density is reached, cells are infected with NDV to initiate the viral production phase. For batch productions, the bioreactor is interrupted for harvest when the known peak production timepoint is reached, typically between 24 and 48 h post infection (hpi) [3,100].

The processes also differ in terms of waste treatment: infected ECEs result in solid waste that is typically incinerated [104], while reusable bioreactors use methods such as cleaning/sterilization in place and chemical treatment of liquid waste, and single-use bioreactors are disposed of through chemical or physical treatments [105].

### 5.2. Parameters for NDV Production in Cell Culture

When using lentogenic NDV strains to infect cell lines for propagation, an exogenous protease must be provided to cleave the F protein and activate infection [28]. This can be achieved by providing allantoic fluid at a concentration from 5 to 10% [20,72,106], or by adding TPCK-treated trypsin [3,28], as is done for influenza [107]. Trypsin is more suitable than allantoic fluid due to the variability regarding undefined animal products in cell culture. This is the same reason why serum-free media is preferred over the use of fetal bovine serum (FBS) in industry [108]. Alternatively, there are strains which can replicate without trypsin addition, including mesogenic strains such as BC and R2B, or strains with a modified F cleavage site, such as LaSota/VF [28]. The optimal trypsin concentration to produce recombinant NDV LaSota in suspension Vero cells was found to be 1 μg/mL [3], although NDV constructs expressing certain antigens can become self-sufficient for viral entry and no longer require trypsin [109].

Another key parameter is the multiplicity of infection (MOI), which has been optimized in suspension Vero cells using serum-free media. Our past research showed that MOIs of 0.1, 0.01 and 0.001 resulted in similar peak production titers (around 10^8^ TCID_50_/mL), while production at an MOI of 0.0001 was approximately 100-fold lower [3]. Replication of NDV in DF-1 and adherent Vero cells is usually achieved with an MOI of 0.01 [58,69,70,72,85,106], which is also within this range. A study using adherent Vero cells with microcarriers in media containing FBS showed similar titers of around 4 × 10^7^ TCID_50_/mL for MOI 2 and 0.2 [100]. When infecting BHK-21 cells, an MOI of 5 resulted in a peak of around 10^7^ EID_50_/mL at 24 hpi, while an MOI of 0.01 caused a delay in that peak to 96 hpi [109]. Thus, NDV seems to have wide range of MOIs that achieve adequate production and the optimal MOI can depend on other process and culture parameters, although 0.01 seems to be a suitable MOI for most conditions.

NDV can be adapted to a cell line by serial passaging, which selects for more efficient replication. This facilitates the next infection and can lead to a higher yield, with an observed increase of around 500-fold in lentogenic strains after four passages in suspension Vero cells and 13-fold in suspension HEK293 [3]. Another study found an increase from 5- to 25-fold in lentogenic strains and from 6- to 10-fold in mesogenic strains after eight passages in adherent Vero cells [39].

Recombinant lentogenic NDV production in several cell lines resulted in titers around 10^7^ and 10^8^ infectious units per mL between 30 and 48 hpi, including suspension Vero cells [3] and adherent cells: Vero [100,106], DF-1 [13,72], BHK-21 [109] and MDCK [41]. The highest titers reported were around 10^9^ PFU/mL in DF-1 [70], 5 × 10^8^ PFU/mL in MDCK [41] and 2.37 × 10^8^ TCID_50_/mL in suspension Vero cells [3]. Production of the mesogenic strain R2B in adherent Vero cells were between 6 × 10^8^ and 6 × 10^7^ TCID_50_/mL [58,85,110]. The production in ECEs is similar but still overall higher, with titers around 10^8^ and 10^9^ infectious units per mL [13,41,54,67,82,109].

### 5.3. NDV Production in Lab-Scale Bioreactors

Although there is still a small number of studies available, a few successful productions of NDV in bioreactors have been published. It is important to note that serum-free media is preferred for industrial bioprocesses, as the use of FBS implies an undefined composition and lot-to-lot variability [108]. In addition, suspension cell lines are preferred over adherent cells, as they are not limited by the surface area available, resulting in a more straightforward scale-up and homogenization of cultures. It is still possible to use adherent cells in a stirred-tank bioreactor using microcarriers, which are beads that the cell can attach to, whereas the microcarriers themselves remain suspended and stirred in the media [111]. Adherent cells can also be used in other bioreactor models, such as fixed-bed and wave, but all methods are ultimately limited by surface area [108].

In 2010, a pioneer work reported the production of the lentogenic F strain in adherent Vero cells using microcarriers in a 2 L bioreactor scale, reaching a peak production of 4.79 × 10^7^ TCID_50_/mL [100]. Although the production was sufficient, the process developed has limited scalability due to the use of serum and adherent cell culture. The same authors tested DF-1 cells in the same microcarrier and stirred-tank bioreactor system, but the resulting titer was low (1.03 × 10^3^ TCID_50_/mL) [102].

In 2021, our group published 1 L bioreactor productions of NDV constructs based on the LaSota strain [3] using a recently developed suspension Vero cell line [112] in commercial serum-free media. Adequate production comparable to ECEs was achieved for all constructs, with a peak titer of 2.37 × 10^8^ TCID_50_/mL for NDV-GFP and 3.16 × 10^7^ TCID_50_/mL for the COVID-19 vaccine candidate NDV-FLS [3]. This work established the basis for the upstream and analytics of a cell culture-based process for NDV production, showing the potential for a scalable process.

### 5.4. Downstream Processing and Formulation

Aside from the upstream production in cell culture, it is also important to establish downstream protocols and formulation. Although it is possible to inject the harvested allantoic fluid from eggs directly into animals for experiments [74], a few studies have implemented sucrose gradient centrifugation to purify NDV [18,27,70,75]. However, this is not the ideal method for industrialization, as it can be impractical and lack reproducibility. Chromatography-based purification methods would be more appropriate for industrial applications due to high scalability and reproducibility [113]. Chromatography purification protocols have been developed for other enveloped viruses, such as lentivirus [114], and could potentially be developed for NDV as well.

Formulation is also an essential aspect that must be studied for NDV. Vaccines with low stability might require storage and transportation in temperatures below −20 °C, which poses a major bottleneck in related to cold chain infrastructure [5]. A few studies have implemented lyophilized versions of their vaccine candidates [27,115], which represents a promising way of simplifying transportation. The lyophilization of a COVID-19 vaccine candidate did not significantly reduce the virus infectivity and allowed for storage at 4 °C [27], which greatly decreased the burden on storage. Further optimization of formulation strategies for NDV and stability studies for the existing vaccine candidates are extremely important to prepare for large-scale manufacturing and distribution.

## 6. Conclusions

In a time where vaccines are in high demand, it is important to establish vaccine platforms with rapid development and scalable manufacturing. NDV is a promising viral vector for vaccines with well-established recombinant technology, a long history of safety in humans and animals, and extensive pre-clinical research. With the progression of clinical trials for NDV-vectored vaccines in humans, it is important more than ever to fill the gap of process development for this virus by testing and optimizing scalable production methods for NDV using cell culture in bioreactors. The basis for a cell culture-based NDV production process has been established, but improvements in the upstream and downstream processing are critical, especially when it comes to purification, formulation and process intensification.

The results achieved in batch bioreactor productions of NDV were comparable to those in ECEs but could potentially be further optimized through process intensification. Different modes of operation, such as fed-batch and perfusion, could reduce by-products and replenish nutrients in the media, potentially allowing cells to reach a higher cell density and higher titers of viral production due to a more favorable metabolic state [19]. Perfusion could be particularly beneficial for NDV, seeing as this virus has been shown to lose infectivity over time in bioreactors [3]. This indicates viral degradation in the bioreactor due to unfavorable temperatures and shear stress from agitation, which could be avoided using perfusion to continuously harvest the virus and reduce the retention time in the vessel, as has been done for the VSV [112]. This can also potentially be integrated with purification for a continuous or semicontinuous manufacturing process [19].

Further development in these aspects is necessary so that NDV-vectored vaccines approved in the future can be produced sufficiently at a large scale without major transportation and cold chain issues.

## Figures and Tables

**Figure 1 viruses-14-00975-f001:**
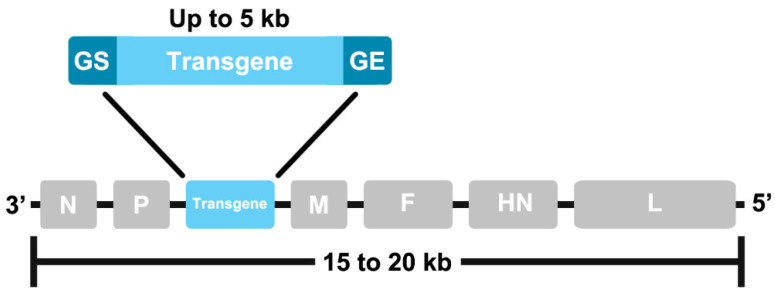
A representation of the NDV single-stranded negative-sense RNA genome including the six native transcriptional units. Transgenes are typically inserted between the P and M genes and flanked by NDV gene start (GS) and gene end (GE) sequences. The total genome length must be an even multiple of six (“rule of six”) to ensure complete encapsidation.

**Figure 2 viruses-14-00975-f002:**
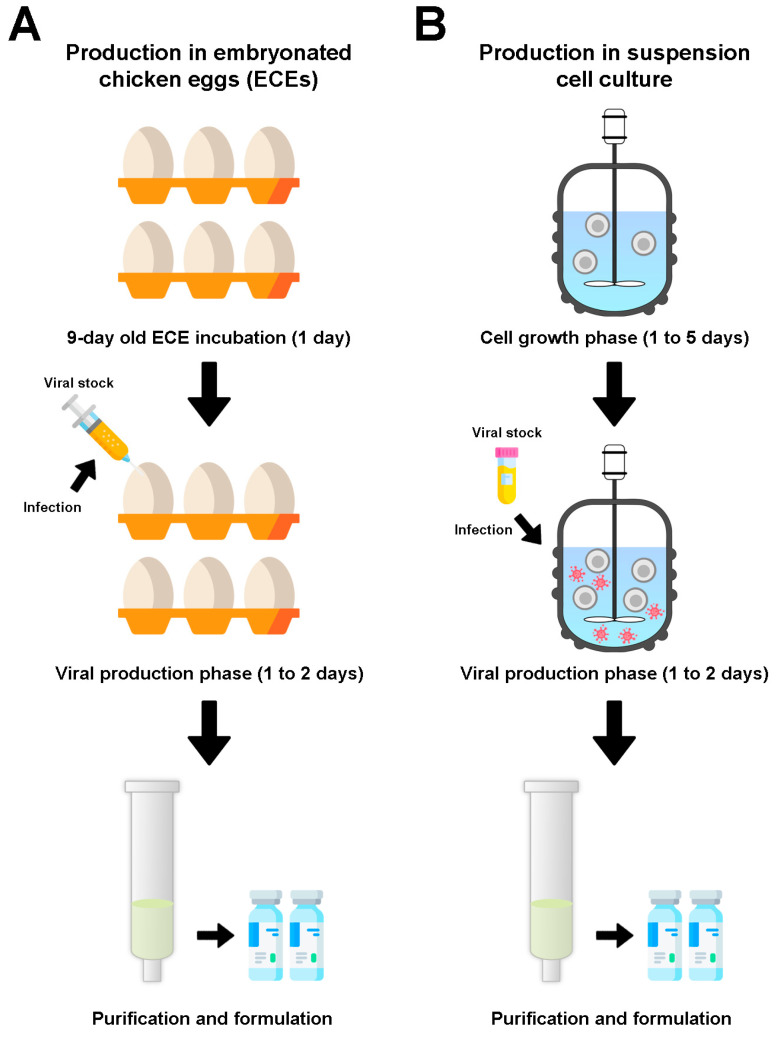
Overview of production processes for viral-vectored vaccines in (**A**) embryonated chicken eggs (ECEs) and (**B**) suspension cell cultures in stirred-tank bioreactors.

**Table 1 viruses-14-00975-t001:** NDV-vectored vaccine candidates for human use in chronological order (publication date).

Vaccine Type	Pathogen	Disease	Antigen	Animal Model	Production Platform *	Route *	Dose *	Reference
Live B1 strain	Influenza A H1N1	Respiratory infection	HA	Mouse	ECEs	iv, ip	two doses; 3 × 10^7^ PFU	[65]
Live LaSota or BC strain	HPIV3	Respiratory infection	HN	African green monkey; rhesus monkey	ECEs, DF-1 cell line	in + it	two doses; 3 × 10^6^ PFU	[20]
Live B1 strain	HRSV	Respiratory infection	F	Mouse	ECEs	in	one dose; 5 × 10^5^ PFU	[66]
Live BC or LaSota/VF strain	SARS-CoV	Respiratory infection	S, S1	African green monkey	ECEs, DF-1 cell line	in + it	one or two doses; 2 × 10^7^ PFU	[28]
Live LaSota strain	HPAIV H5N1	Respiratory infection	HA	African green monkey	DF-1 cell line	in	two doses; 2 × 10^7^ PFU	[40]
Live B1 strain	HIV	AIDS	Gag	Mouse	ECEs, Vero cell line	in	two doses; 5 × 10^5^ PFU prime and 10^6^ PFU boost	[67]
Live LaSota or BC strain	EBOV	Ebola virus disease (hemorrhagic fever)	GP	Rhesus monkey	DF-1 cell line	in + it	two doses; 10^7^ PFU	[29]
Live LaSota strain	HIV	AIDS	Gag	Mouse	ECEs	in	two doses; 5 × 10^5^ PFU prime and 10^6^ PFU boost	[68]
Live LaSota/VF strain	*Borrelia burgdorferi*	Lyme	BmpA, OspC	Hamster	ECEs	in, im, ip	one or two doses; 10^6^ PFU	[69]
Live LaSota strain	HIV	AIDS	Env (gp160)	Guinea pig	ECEs	in, im, in + im	two or three doses; 3 × 10^5^ PFU (in) or 5 × 10^5^ PFU (im)	[70]
Live LaSota strain	NiV	Encephalitis	F, G	Mouse, pig	ECEs	im	two doses; 10^8^ EID_50_ (mice) or 2 × 10^9^ EID_50_ (pigs)	[71]
Live LaSota or Lasota/BC strain	NoV	Gastroenteritis	VP1	Mouse	ECEs	in	one dose; 10^6^ EID_50_	[72]
Live LaSota strain	HIV	AIDS	Env (gp160)/Gag (p55)	Guinea pig, mouse	ECEs	in	two doses; 2 × 10^5^ PFU (guinea pigs) or 4 × 10^4^ PFU (mice)	[50]
Live LaSota strain	HIV	AIDS	Env (gp160)	Guinea pig	ECEs	in	four doses of 2 × 10^5^ PFU or two doses of 2 × 10^5^ PFU followed by two doses of recombinant protein (gp120 or gp140)	[73]
Live LaSota strain	Poliovirus	Poliomyelitis	P1/3CD	Guinea pig	ECEs	in	two doses; 10^5^ PFU	[49]
Live chimeric NDV strain	EBOV	Ebola virus disease (hemorrhagic fever)	GP	Guinea pig	ECEs	in	two doses; 2 × 10^6^ TCID_50_	[13]
Live LaSota strain	JEV	Encephalitis	E, NS1	Mouse	ECEs	in	one dose; 10^6^ EID_50_	[74]
Live LaSota strain	SARS-CoV-2	COVID-19	S, S-F chimera	Mouse	ECEs	in	two doses; 10 or 50 μg	[18]
Inactivated LaSota strain	SARS-CoV-2	COVID-19	S, S-F chimera	Mouse, hamster	ECEs	im	two doses; 5 or 10 μg	[75]
Live B1 strain	SARS-CoV-2	COVID-19	S	Mouse, hamster	ECEs	in	one or two doses; 10^4^ PFU (mice) or 10^6^ PFU (hamster)	[26]
Live or inactivated LaSota strain	SARS-CoV-2	COVID-19	HXP-S	Pig	ECEs	in, im, in + im	two doses of 10^7^, 3 × 10^7^, 10^8^ or 3 × 10^8^ EID_50_ (live); two doses of 10^8^ EID_50_ (inactivated)	[31]
Live or inactivated LaSota strain	SARS-CoV-2	COVID-19	HXP-S	Hamster, mouse	ECEs	in, im, in + im	two doses of 1.0, 0.3, 0.1, 0.03 or 0.03 μg (im, inactivated, hamsters); two doses of 10^6^ EID_50_ (in, live, hamsters); two doses of 10^4^, 10^5^ or 10^6^ EID_50_ (in prime and im boost, live, mice); two doses of 1 μg (im, inactivated, mice)	[76]
Live LaSota strain	SARS-CoV-2	COVID-19	HXP-S	Rat	ECEs	in, im, in + im	two doses; 7.4 × 10^8^ EID_50_	[30]
Live LaSota strain	SARS-CoV-2	COVID-19	S, truncated S	Hamster	ECEs	in	one or two doses; 10^7^ PFU	[27]
Live LaSota strain	EBOV	Ebola virus disease (hemorrhagic fever)	GP	Mouse	ECEs	in	two doses of 10^6^ PFU; one dose of 10^6^ PFU and one dose of adenovirus-vectored vaccine	[77]
Inactivated VG/GA strain	SARS-CoV-2	COVID-19	RBD	Mouse	ECEs	im	two doses of 1, 5 or 10 μg	[78]

* ECEs: embryonated chicken eggs, iv: intravenous, ip: intraperitoneal, in: intranasal, it: intratracheal, im: intramuscular, PFU: plaque-forming units, TCID_50_: tissue culture infectious dose 50%, EID_50_: embryo infectious dose 50%.

**Table 2 viruses-14-00975-t002:** Recombinant NDV-vectored vaccine candidate in clinical trials for human use.

Responsible Group	Vaccine Type	Pathogen	Disease	Antigen	Phase	Route *	Dose *	Reference
Icahn School of Medicine at Mount Sinai, USA	Live LaSota strain	SARS-CoV-2	COVID-19	HXP-S	I	in, im, in + im	3.3 × 10^8^ EID_50_ 1 × 10^9^ EID_50_	NCT05181709
Laboratorio Avi-Mex, Mexico	Live LaSota strain	SARS-CoV-2	COVID-19	HXP-S	I/II	im, in + im	10^8^ EID_50_	NCT04871737 [79] NCT05205746
Institute of Vaccines and Medical Biologicals, Vietnam	Inactivated LaSota strain	SARS-CoV-2	COVID-19	HXP-S	I/II	im	1, 3 or 10 μg	NCT04830800
Butantan Institute, Brazil	Inactivated LaSota strain	SARS-CoV-2	COVID-19	HXP-S	I/II	im	1, 3 or 10 μg	NCT04993209
Mahidol University, Thailand	Inactivated LaSota strain	SARS-CoV-2	COVID-19	HXP-S	I/II	im	1, 3 or 10 μg	NCT04764422 [80]

* in: intranasal, im: intramuscular, EID_50_: embryo infectious dose 50%.

## Data Availability

Not applicable.

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
