# Peer review of "Development and Scalable Production of Newcastle Disease Virus-Vectored Vaccines for Human and Veterinary Use"

_viruses, 2022, doi:10.3390/v14050975_

Round 1

Reviewer 1 Report

The manuscript entitled “Development and scalable production of Newcastle Disease Virus-vectored vaccines for human and veterinary use” describes the current development in NDV-vectored vaccines for human and animal uses and elaborates their production system for large scale manufacturing. The manuscript is interesting, well written and included the latest development in the field of NDV vectored vaccines for human and veterinary uses. The author could include some of the suggestions mentioned below while revising the manuscript.

  1. Under the section 3, (Designing and generating recombinant NDV), the author could add some details of the cloning and expression strategies, for example, the relative location of the transgene insertion, the GS and GE sequences, rule of six, etc.
  2. Table 1: There are a number of published reviews that summarized the list of NDV-vectored vaccine candidates for human and veterinary uses. It would be useful, if the author also could include the current status of development of these vaccines listed in the table (if the information are available).
  3. Under the section 4.1. (Vaccines for human use), the author could focus some of the potential NDV-vectored vaccine candidates for human uses, in particular against the SARS-CoV-2 or Ebola, and discuss their immunogenicity and protection in animal models, field trials data (if available). This would strengthen and highlight the potentials of NDV as a vector against the pandemic human diseases.

Author Response

We would like to thank the reviewer for his time and valuable comments. Please find below a point-by-point response to the reviewers' comments.

Comment 1: Under the section 3, (Designing and generating recombinant NDV), the author could add some details of the cloning and expression strategies, for example, the relative location of the transgene insertion, the GS and GE sequences, rule of six, etc.

Response: Additional details were added on lines 128-132 regarding GS and GE sequences and on line 139 regarding the location of transgene insertion. The caption of Figure 1 was modified to mention the rule of six (lines 136-137). Moreover, lines 138-143 elaborate on the relative location of transgene insertion and the rule of six and all this information is shown in Figure 1 and its caption.

Comment 2: Table 1: There are a number of published reviews that summarized the list of NDV-vectored vaccine candidates for human and veterinary uses. It would be useful, if the author also could include the current status of development of these vaccines listed in the table (if the information are available).

Response: This is a great suggestion. Out of all the vaccine candidates listed in Table 1, the only ones found to have advanced to clinical trials are listed in Table 2. As such, we added a sentence to clarify the status of development of the vaccine candidates (lines 234-235), but we opted not to include this information in the Table 1, as it might overcrowd the table with information that is now expressed in the text.

Comment 3: Under the section 4.1. (Vaccines for human use), the author could focus some of the potential NDV-vectored vaccine candidates for human uses, in particular against the SARS-CoV-2 or Ebola, and discuss their immunogenicity and protection in animal models, field trials data (if available). This would strengthen and highlight the potentials of NDV as a vector against the pandemic human diseases.

Reponse: Content was added on lines 214-233 to elaborate on immunogenicity in animal models in studies focusing on EBOV, SARS-CoV and SARS-CoV-2. A paragraph was also added on lines 252-264 discussing the clinical data that has been published regarding the NDV-vectored SARS-CoV-2 vaccines.

Reviewer 2 Report

Julia P. C. Fulber and Amine A. Kamen submitted a manuscript titled with “Development and scalable production of Newcastle Disease Virus-vectored vaccines for human and veterinary use” to viruses. Although this review is short and not contains all parts about NDV vector vaccine, it supplies some contents comparing other reviews, especially about NDV production in cell culture and SARS-CoV2 vaccine based on NDV. This review organizes and discuss well. I suggest it can be accepted with minor modification:

Line 65-70: discuss advantage of cell culture in terms of treatment of waste during NDV production.

Line 84: add this recent reference “Acute Keratoconjunctivitis Resulting From Coinfection With Avian Newcastle Virus and Human Adenovirus”.

Line 127-130:  discuss about UTR, including GS and GE, control expression of foreign gene.

Line 210-224: Although NDV has been used as vector to express antigen of human pathogens, but, according to my knowledge, no vaccine is in clinical trials, except of SARS-CoV2. Thus, if authors show more contents about results of clinical trials of SARS-CoV2 vaccine based on NDV vector, it will be more interesting.  

Author Response

We would like to thank the reviewer for his time and valuable comments. Please find below a point-by-point response to the reviewers' comments.

Comment 1: Line 65-70: discuss advantage of cell culture in terms of treatment of waste during NDV production.

Response: Information about the differences in waste treatment between cell culture and egg-based processes was added to lines 323-326 in section 5 “Manufacturing of NDV-vectored vaccines”.

Comment 2: Line 84: add this recent reference “Acute Keratoconjunctivitis Resulting From Coinfection With Avian Newcastle Virus and Human Adenovirus”.

Response: The reference has been added on Line 84.

Comment 3: Line 127-130:  discuss about UTR, including GS and GE, control expression of foreign gene.

Response: Information about the use of different UTRs to control expression levels of foreign genes has been added in lines 127-132.

Comment 4: Line 210-224: Although NDV has been used as vector to express antigen of human pathogens, but, according to my knowledge, no vaccine is in clinical trials, except of SARS-CoV2. Thus, if authors show more contents about results of clinical trials of SARS-CoV2 vaccine based on NDV vector, it will be more interesting.  

Response: A paragraph has been added (lines 252-264) summarizing the results of the clinical trials for the NDV-vectored SARS-CoV-2 vaccine candidates.

Reviewer 3 Report

The authors bring in this review information about the use of NDV as a viral vector, describe the steps and limitations to generating recombinant NDV constructs, showing the advances in human and veterinary vaccine candidates in pre-clinical and clinical tests. The authors also discuss the existing data on NDV propagation and provide prospects for the potential steps necessary to potentially achieve large-scale NDV-vectored vaccine manufacturing 

Author Response

We would like to thank the reviewer for his time and valuable comments.